# A digital twin for DNA data storage based on comprehensive quantification of errors and biases

Andreas L. Gimpel [1], Wendelin J. Stark[1], Reinhard Heckel[2] & Robert N. Grass [1] ✉

Archiving data in synthetic DNA offers unprecedented storage density and longevity. Handling and storage introduce errors and biases into DNA-based storage systems, necessitating the use of Error Correction Coding (ECC) which comes at the cost of added redundancy. However, insufficient data on these errors and biases, as well as a lack of modeling tools, limit data-driven ECC development and experimental design. In this study, we present a comprehensive characterisation of the error sources and biases present in the most common DNA data storage workflows, including commercial DNA synthesis, PCR, decay by accelerated aging, and sequencing-by-synthesis. Using the data from 40 sequencing experiments, we build a digital twin of the DNA data storage process, capable of simulating state-of-the-art workflows and reproducing their experimental results. We showcase the digital twin's ability to replace experiments and rationalize the design of redundancy in two case studies, highlighting opportunities for tangible cost savings and data-driven ECC development.

As the amount of digital data to be stored continues to grow by Zettabytes every year, DNA is considered a potential alternative to conventional storage media due to its exceptional stability and storage density[1–3]. The use of DNA as a storage medium presents unique practical challenges, such as affordability and scalability, as well as design challenges, such as the choice of redundancy and algorithm for error correction coding (ECC)[3–5]. The latter challenge is aggravated by the errors incurred by data stored in DNA, ranging from single-site errors (i.e., substitutions, deletions, and insertions) to sequence dropout (i.e., the loss of data-encoding sequences)[4]. While errors stem directly from the chemical or biological processes involved in the DNA data storage workflow (e.g., synthesis, amplification, aging, and sequencing), sequence dropout is the product of a biased distribution for the oligonucleotide count per sequence (i.e., the coverage distribution). Due to these errors and biases, data stored in DNA is encoded with redundancy using ECC[4,6,7]. These coding schemes add redundancy to recover the encoded data from the DNA sequences

while correcting a limited number of errors and tolerating some missing sequences. In practice, choosing the optimal level and type of redundancy requires a priori knowledge of the expected error and dropout rates, for which insufficient information is often available. Due to this lack of understanding of how experimental choices, from synthesis provider to storage workflow, affect the expected error and dropout rates, experience, and overcompensation currently often guide the choice of redundancy levels.

Beyond just choosing an adequate redundancy level, choosing a suitable ECC from the many implementations reported to date[6,8–11] requires standardized error scenarios facilitating meaningful and fair comparisons. Computational comparisons have relied on fictitious error scenarios[10,11]—considering error types in isolation—while experimental comparisons are costly and potentially misleading due to the plethora of potentially critical experimental parameters. In silico tools for the simulation of errors in DNA exist[12–14], but they often do not support the parallel simulation of large oligonucleotide pools, neglect

[1]Department of Chemistry and Applied Biosciences, ETH Zürich, Vladimir-Prelog-Weg 1-5, 8093 Zürich, Switzerland. [2]Department of Computer Engineering, Technical University of Munich, Arcisstrasse 21, 80333 Munich, Germany. ✉e-mail: robert.grass@chem.ethz.ch

sequence dropout due to evolving bias in the coverage distribution, or directly reproduce experimental error patterns without considering experimental parameters. To replace experiments or compare ECCs however, an in-silico tool for DNA data storage must accurately reflect the errors and sequence dropout of state-of-the-art workflows based only on experimental parameters. This requires a systematic understanding of the individual sources of errors and biases encountered in such workflows.

Many of the biological and synthetic methods used in common DNA data storage workflows are well characterized (e.g., oligonucleotide synthesis[15,16], PCR[17,18], sequencing-by-synthesis (SBS)[19,20]). In contrast, studies on DNA data storage often only quantify overall error rates−if at all−and do not consider coverage biases. The works by Heckel et al.[4] and Chen et al.[21] began quantifying these error sources in isolation, identifying significant biases related to the synthesis and amplification of oligonucleotide pools. Still, no study has systematically investigated the evolution of error rates and coverage biases throughout the entire DNA data storage workflow.

In this work, we comprehensively characterize the error sources and biases present in the most widely used DNA data storage workflows to date[1,7]. This includes commercial DNA synthesis from the two major providers of large-scale oligonucleotide pools used in the literature[1] (i.e., Twist Biosciences and Genscript/CustomArray), amplification via PCR, long-term storage and decay by accelerated aging, and sequencing by Illumina's SBS technology. For our investigation, we systematically sequenced oligonucleotide pools throughout the workflows to analyze their error profiles and coverage distributions, for a total of 40 sequencing datasets. By characterizing the base preferences, positional dependencies, and distributional inhomogeneities of all errors, we provide a complete description of all error sources in the various steps of the workflows. In addition, the analysis of coverage distributions revealed any potential coverage bias from synthesis, amplification, and aging, which we show to be critical for understanding sequence dropout. Finally, we condense the data on error rates and biases into a digital twin of the DNA data storage process: a tool to explore experimental workflows and provide standardized simulations for experimental scenarios. We demonstrate the digital twin's ability to reproduce state-of-the-art workflows and showcase its application to the data-driven design of redundancy, which offers opportunities to replace costly experiments and facilitate meaningful comparisons between ECCs.

Lastly, while upcoming synthesis (e.g., photochemical or enzymatic synthesis) and sequencing processes (e.g., nanopore sequencing)−which are not yet widely adopted for DNA data storage−are not considered here, our characterization workflow and digital twin provide a suitable blueprint to analyze these emerging technologies in the future.

## Results

In this work, we characterize errors and biases from sequencing data using four oligonucleotide pools, each with 12000−12472 sequences of 143-157 nucleotides (nt). Two pools were synthesized via an electrode array-based method (Genscript/CustomArray) and two by a material deposition-based technology (Twist Biosciences). All pools consisted of random sequences, with one pool each enforcing a constraint on GC content of 50% ("GC-constrained"), while the other remained unconstrained (see Methods and Supplementary Table 1). All pools were used in two workflows, consisting of either extensive re-amplification with up to 90 PCR cycles or accelerated aging up to an equivalent storage duration of 1000 years at 10 °C. Throughout the process, samples of the pools were sequenced to track the evolution of errors and biases for a total of 40 experimental endpoints across the two workflows. For our analysis, errors and biases were characterized by aligning sequencing reads to their respective references, identifying mutations,

and evaluating the resulting error patterns. For more details on the analysis procedure and the datasets used, we refer to the Methods and Supplementary Note 1.

In the following, we first quantify the overall error rates in our experiments, followed by the characterization of each individual error source in the data storage workflow. We then build and verify a computational model of the workflow, which is used in a case study to illustrate its value for the data-driven choice of redundancy in ECCs.

### Identifying error sources and assessing error independence

To validate our experimental approach, we first compared our overall error rates to those published in previous studies. Throughout all our 40 datasets, we observed overall error rates of $6.7 \pm 6.9$ deletions, $7.9 \pm 2.0$ substitutions, and $<0.3 \pm 0.2$ insertions per thousand nucleotides (i.e., $10^{-3}$ nt$^{-1}$) on average, in-line with error rates published in other studies[4,22,23]. Variation in the observed deletion and substitution rates between different experimental conditions and different oligonucleotide pools was large, with maximum rates of $17.1 \times 10^{-3}$ nt$^{-1}$ deletions and $12.5 \times 10^{-3}$ nt$^{-1}$ substitutions, respectively. Analyzing the variance across the measured error rates in this diverse dataset (three-way ANOVA with HC3 correction, see Fig. 1a)−considering synthesis provider, number of PCR cycles, and storage duration as factors in a main effects analysis−showed that synthesis and PCR were the major error sources in our experiments. The synthesis process explains most of the difference observed in deletion rates ($F(1, 76) = 933.7$, $p = 10^{-44}$), accounting for 92% of its variance. This highlights synthesis as a dominating source of deletions, as noted by others[15,16], and identifies a large difference in fidelity between synthesis processes. In contrast, substitution rates varied most between samples with different sample preparations. PCR was found to be the main factor affecting substitutions ($F(1, 76) = 1251$, $p = 10^{-49}$), accounting for 86% of the variance (see Fig. 1a). The full ANOVA results are presented in Supplementary Table 8.

Next, we assessed error independence in our datasets, i.e., the assumption that mutations occur independently from one to another, which is often inherently assumed when modeling errors in DNA[10,11,13]. To do so, we compared the frequency distributions of consecutive errors and errors per read to those expected assuming that errors are introduced independently. Under error independence, we expect to observe consecutive errors according to a geometric distribution with a success probability equal to the average error rate. We found that, while the frequency of consecutive substitutions closely matches its theoretical distribution (see Fig. 1c), the occurrence of multiple consecutive deletions was considerably more frequent (see Fig. 1b). Runs of consecutive deletions−with a mean length of 2.6 bases and referred to as a deletion event−were overrepresented and accounted for 10−14% of all deletions, depending on the synthesis process. Going further, the frequency distribution of errors per read is expected to be binomially distributed under the assumption of error independence, with the length of the sequence and the average error rate as parameters. Substitutions showed good agreement to this theoretical distribution (see Fig. 1e), whereas deletion events behaved differently depending on synthesis technology (see Fig. 1d). For electrochemical synthesis, deletion events were heavily clustered in a small subset of reads. While this led to a greater proportion of deletion-free reads (52% vs. 35% expected) and a small number of reads with only one or two deletions (35% vs. 56% expected), about 13% (vs. 9% expected) of oligonucleotides in these pools featured at least three deletions. No clustering across reads was evident for the material deposition-based synthesis, as deletions were generally rare. Taken together, this analysis established that the assumption of error independence is generally valid for substitutions, but is violated for deletions, which tend to cluster both within and across reads in the electrochemical synthesis.

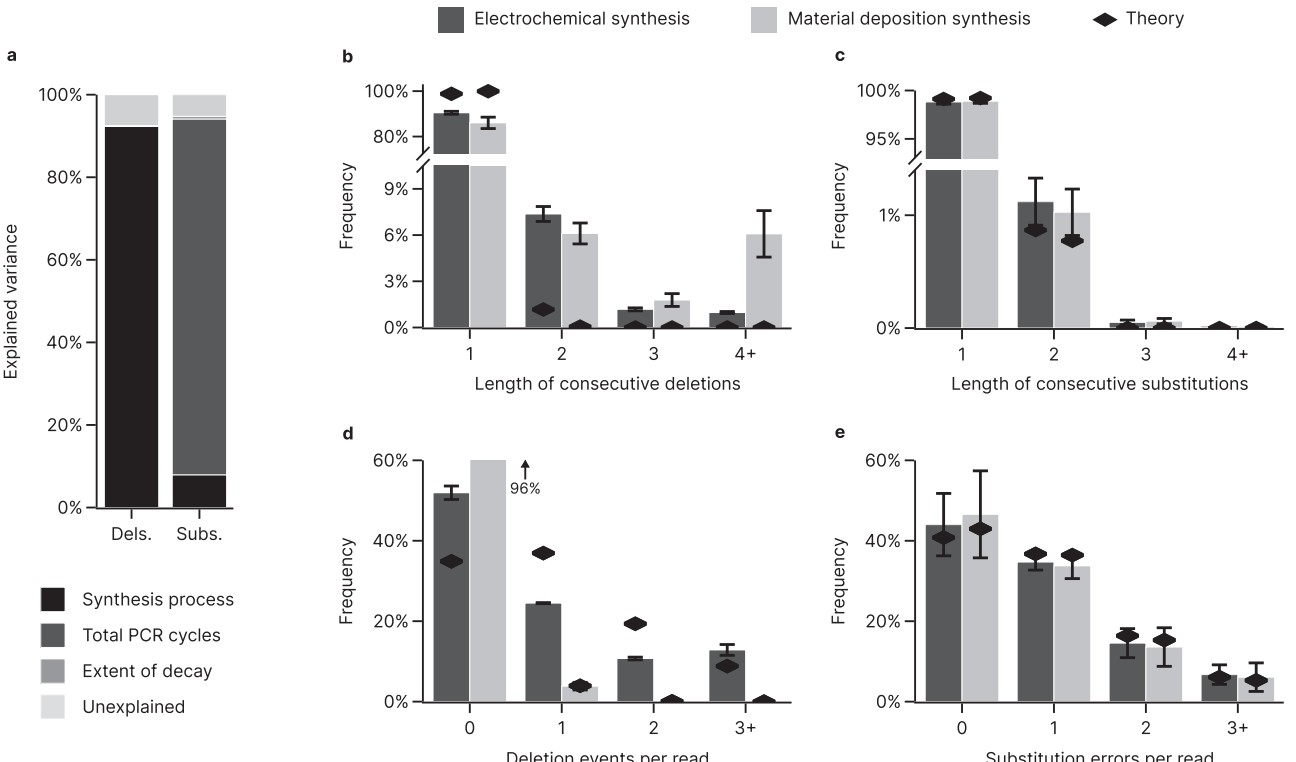

**Fig. 1 | Overview of error variance and general error distributions. a** The contributions of the synthesis process, PCR cycles, and extent of decay to the overall variance in mean deletion (Dels., left) and substitution (Subs., right) rates between samples were assessed by four-way analysis of variance (ANOVA, see Methods and Supplementary Table 8). **b**–**e** Distributional analysis of error independence for deletions (**b**, **d**) and substitutions (**c**, **e**) based on the observed frequency of error runs (**b**, **c**) and errors per read (**d**, **e**). Bars show the mean frequency for the GC-unrestricted pools synthesized by electrochemical (dark gray, $n = 8$ sequencing experiments, with both forward and reverse reads used) and material deposition (light gray, $n = 11$ sequencing experiments, with both forward and reverse reads used) processes, with error bars indicating the standard deviation of the sample. Theoretical distributions expected under the assumption of error independence are also shown (black diamonds), representing a geometric distribution parameterized by the mean error rate for consecutive errors (**b**, **c**) and a binomial distribution parameterized by the length of the sequence and the mean error rate for the errors per read (**d**, **e**, see also Methods). The histogram for deletions per read treats any run of deletions as a single event to accommodate the non-ideality of deletion runs. Source data are provided as a Source Data file.

## Not all DNA is created equal: synthesis errors and coverage biases

As noted above, the large difference in mean deletion rate between electrochemical ($13.5 \pm 2.0 \times 10^{-3}$ nt$^{-1}$) and material deposition-based ($0.58 \pm 0.15 \times 10^{-3}$ nt$^{-1}$) synthesis identified synthesis as the main error source for deletions. This is corroborated by the positional dependence of deletions in the sequencing reads, which showed a distinct increase in the synthesis direction for the electrochemical synthesis (i.e., 3′–5′ for the forward read, 5′–3′ for the reverse read, Fig. 2a). The strongly increasing deletion rate observed towards the 5′-end of the electrochemically synthesized oligonucleotides, >5% per nucleotide, likely stems from mass transfer limitations. As the synthesized oligonucleotide becomes longer, the distance to the acid-generating electrode grows and steric hindrance increases the electrochemical cell resistance, impeding acid-induced deprotection and preventing both subsequent addition of the next nucleotide and blocking of the erroneous oligonucleotide by capping[24,25]. This also explains the observed deviation from statistical independence for deletions noted previously: oligonucleotides that have already suffered from mass transfer-induced deletions are more likely to do so again in subsequent deprotection steps, leading to a cluster of deletions. Material deposition-based synthesis on the other hand exhibited neither a high deletion rate nor any considerable positional dependence. With a fidelity exceeding one deletion error in 2000 nucleotides, these amplified oligonucleotides were essentially error-free for the purposes of DNA data storage. Despite this large difference in deletion rates, both synthesis processes find broad application in DNA data storage[1],

likely due to considerations of scalability and cost. For both synthesis processes, deletions also did not show any relevant bias towards any nucleotide, and only a negligible number of substitutions were introduced (see Supplementary Note 3 and Supplementary Figure 12).

Focussing on the coverage distributions of the oligonucleotide pools after synthesis, we compared sequencing data obtained after minimal sample preparation (15 PCR cycles and size selection by agarose gel electrophoresis). Similar to other studies[6,21], the normalized coverage distributions of all oligonucleotide pools in our study were positively skewed—featuring a long tail of few sequences at high coverages—and were well approximated by lognormal distributions (see Fig. 2b). Quantifying this coverage bias with the standard deviation of the corresponding lognormal distribution ($\sigma$) highlighted the severe effects of the GC-constraint on the electrochemically synthesized pools. While synthesis by material deposition yielded near-gaussian coverage both with unconstrained and GC-constrained sequences ($\sigma = 0.27$ vs. $\sigma = 0.30$), electrochemical synthesis yielded slightly biased coverage with GC-constrained sequences ($\sigma = 0.58$), and severe bias without constraints ($\sigma = 1.30$, see Fig. 2b). Combined with the significant difference in mean deletion rates between these synthesis methods, the choice of synthesis provider critically affects the baseline error level and coverage bias for DNA data storage.

## Quantifying substitutions and bias introduced via PCR

Generally, PCR introduces both substitution errors and biases into oligonucleotide pools, mainly due to the limited fidelity of the polymerase[4,21]. Previous studies have characterized PCR errors in the

**a**

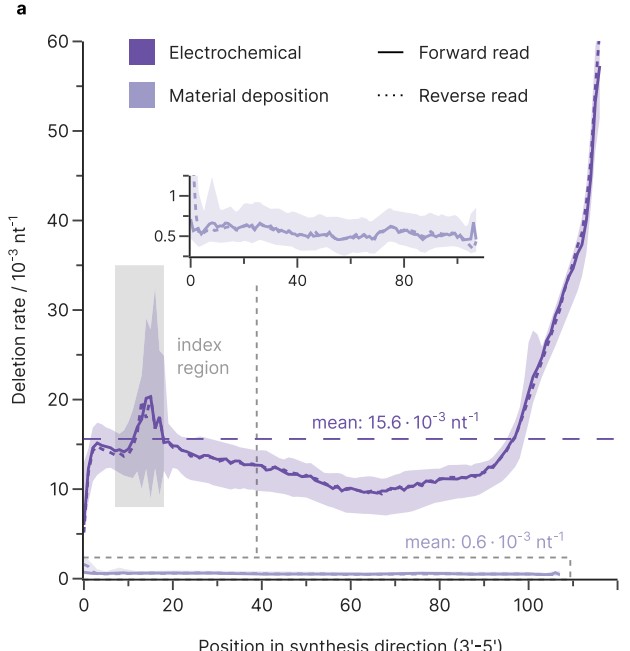

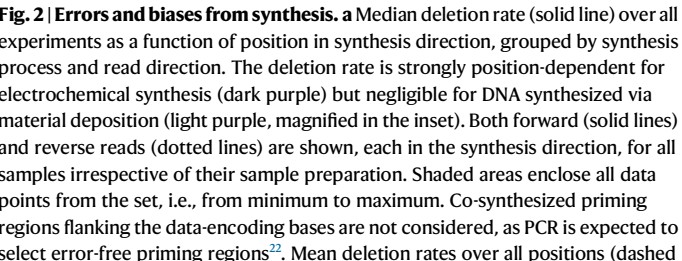

**b**

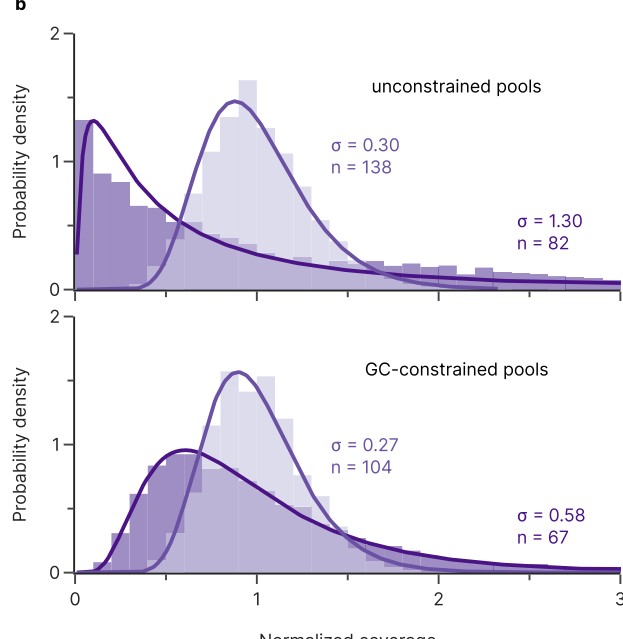

**Fig. 2 | Errors and biases from synthesis. a** Median deletion rate (solid line) over all experiments as a function of position in synthesis direction, grouped by synthesis process and read direction. The deletion rate is strongly position-dependent for electrochemical synthesis (dark purple) but negligible for DNA synthesized via material deposition (light purple, magnified in the inset). Both forward (solid lines) and reverse reads (dotted lines) are shown, each in the synthesis direction, for all samples irrespective of their sample preparation. Shaded areas enclose all data points from the set, i.e., from minimum to maximum. Co-synthesized priming regions flanking the data-encoding bases are not considered, as PCR is expected to select error-free priming regions[22]. Mean deletion rates over all positions (dashed line) and the indexing region (shaded in gray), where the sequences have very low diversity, are also shown. **b** Coverage distributions normalized to the mean sequencing coverage (given as *n*) for oligonucleotide pools with (bottom) and without (top) constraints on GC content from electrochemical (dark purple) and material deposition-based synthesis (light purple) after 15 PCR cycles. All pools fit a lognormal distribution (solid line), but the material deposition-based pools show more even oligonucleotide coverage for both pool types. Standard deviations of the fitted lognormal distributions, *σ*, are shown in the plot. Source data are provided as a Source Data file.

context of genomic sample amplification (e.g., for mutation detection via high-throughput sequencing)[17,18], but PCR errors are also relevant for DNA data storage, where they reduce the fraction of error-free oligonucleotides. To assess this, we characterized the errors introduced during PCR by amplifying samples of the oligonucleotide pools with varying numbers of PCR cycles and quantifying the evolution in error rates (see Fig. 3a). All PCR experiments were stopped well before reaching the plateau phase to ensure an excess of primers and nucleotides for exponential amplification. Sequencing data showed that PCR introduced only substitutions, at a mean rate of $1.09 \times 10^{-4}$ nt$^{-1}$ cycle$^{-1}$ for our Taq-based polymerase (KAPA SYBR FAST), see Fig. 3b and c. The polymerase exhibited a strong bias towards A→G/T→C transitions (61% of substitutions), with further preference for A→T/T→A transversions (13%). This is in-line with the studies quantifying polymerase fidelity based on single amplicons, which found substitution rates within $1 \times 10^{-5}$ to $2 \times 10^{-4}$ nt$^{-1}$ cycle$^{-1}$ for Taq-polymerase, and similar substitution patterns[17,18,26]. Consequently, the established polymerase fidelity metric (i.e., polymerase fidelity relative to Taq-polymerase) can be used to extrapolate the substitution rates expected from other commonly used polymerases in the context of DNA data storage[17,18]. The C→T/G→A transition was also relevant in our experiments (19% of substitutions), but is thought to occur due to temperature-induced cytosine deamination during thermocycling rather than polymerase errors[18].

Stochastic effects of PCR and non-uniform amplification lead to biases in coverage distributions[4,21,27–29]. To quantify this amplification bias in a DNA data storage context, we characterized the distribution of normalized amplification efficiencies, i.e., the ratio $\frac{1+\epsilon_i}{1+\bar{\epsilon}}$ between an individual sequence's efficiency, $\epsilon_i$, and the pool's mean efficiency, $\bar{\epsilon}$, for our datasets. Here, the individual sequence efficiency, $\epsilon_i \in [0,1]$,

represents the probability of successful amplification for each copy of sequence $i$ during one PCR cycle. Assuming negligible stochastic effects (i.e., at high initial coverage), the relative amplification efficiency is related to the experimentally observed fold change in normalized sequence coverage, $x_i$, from sequencing before and after amplification with $c$ cycles, as shown in Eq. 1[29].

$$\frac{1+\epsilon_i}{1+\bar{\epsilon}} = \left(\frac{x_i(c)}{x_i(0)}\right)^{\frac{1}{c}}. \tag{1}$$

We found that the relative amplification efficiencies are normally distributed in our material deposition-based oligonucleotide pools, with a standard deviation of 0.0051 (unconstrained pool) and 0.0048 (GC-constrained pool), see Fig. 3d and Supplementary Fig. 13. To validate our estimate of the overall PCR bias, we replicated this analysis for the sequencing data reported by Chen et al.[21] (change of 31 PCR cycles), Erlich et al.[6] (90 cycles), and Koch et al.[23] (60 cycles). We found amplification biases that were larger, but comparable to ours (see Fig. 3d), with standard deviations ranging from 0.0058 to 0.012. Given these datasets, the broadness of the efficiency distribution does not appear to directly depend on GC constraints and is thus likely caused by experimental conditions. To this end, factors such as the choice of primer, the temperature, and duration of the steps, or the polymerase itself are known to affect amplification efficiency and thus amplification bias, amongst others[30–33]. Specifically the use of high-fidelity, proofreading polymerases (such as by Erlich et al.[6] and Chen et al.[21]), which stall DNA synthesis upon reading uracil, might incur a stronger amplification bias due to cytosine deamination to uracil during storage[34]. Moreover, the repeated dilutions needed after each amplification, albeit performed at high physical coverage, may introduce

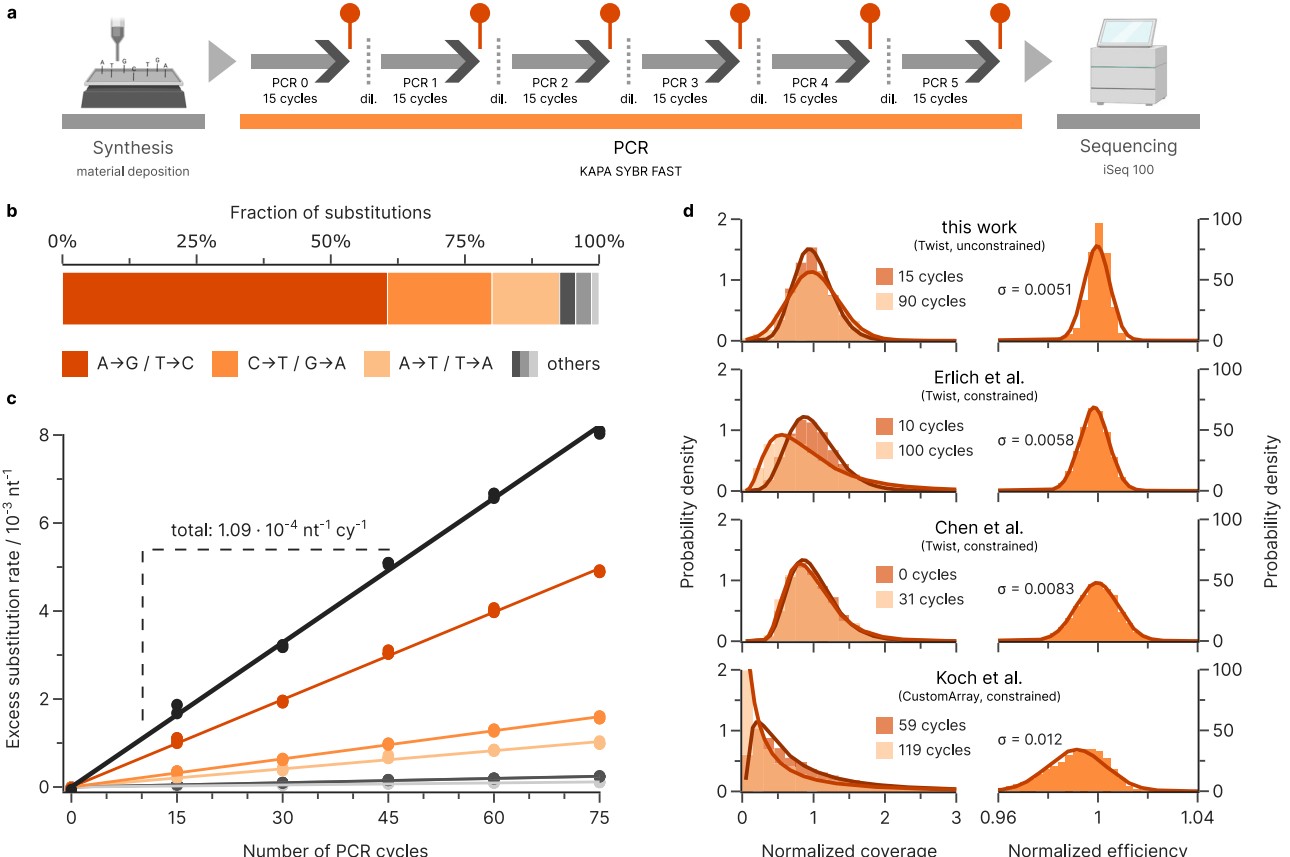

**Fig. 3 | Errors and biases from PCR. a** Experimental workflow for estimating the error rates and biases during PCR. **b**, **c** Substitutions introduced as a function of the number of additional PCR cycles for the oligonucleotide pools from material deposition-based synthesis, using the substitution rate at 15 cycles as the baseline. The regression slope (solid lines) yields an overall error rate of $1.09 \times 10^{-4}$ per nucleotide per cycle ($nt^{-1}$ $cy^{-1}$) and shows A→G/T→C transitions account for 61% of substitutions, followed by C→T/G→A transitions (20%) and A→T/T→A transversions (13%). **d** The normalized coverage distributions (left) of sequencing pools are shown before (dark orange) and after repeated amplification (light orange). Any difference between the pre-PCR and post-PCR coverage distributions can be attributed to PCR bias and/or stochastic effects. Relating the change in coverage pre- and post-PCR to the number of PCR cycles on the sequence level yields an estimate of the efficiency relative to the pool (right). The broadness of the resulting efficiency distribution, characterized by the standard deviation of the fitted normal distributions given in the plots (solid lines), can be interpreted as an upper bound on the overall PCR bias. Comparison shown of efficiency distributions between our experiments, the deep amplification performed by Erlich et al.[6], the bias experiment by Chen et al.[21], and the bunny experiments by Koch et al.[23]. Individual sequences with <10 reads in the sequencing data were removed from this analysis, due to the large uncertainty associated with sampling at low coverage. Source data are provided as a Source Data file.

stochastic effects. The data by Koch et al.[23] is an extreme example of this: after amplification, the DNA was incorporated into silica nanoparticles embedded in polymer involving many handling and dilution steps. In contrast, all experiments of the present study were conducted at high physical coverages (>1000) with sufficient sequencing coverage (>50) to circumvent such stochastic effects. However, as these confounding factors increase the experimentally observed bias, the empirical distributions of the relative amplification efficiencies shown in Fig. 3d can be interpreted as an upper bound on the bias caused directly by PCR amplification in typical DNA data storage experiments.

Due to the exponential nature of PCR, the normally distributed amplification efficiency leads to a progressively more positively skewed coverage distribution with a long tail (see Fig. 3d). This initially small effect thus gains relevance as many amplifications are performed, in-line with observations in literature[28,35]. Considering that data storage workflows routinely use >60 PCR cycles and pools might already be highly skewed from synthesis (see Fig. 2b), PCR considerably biases the oligonucleotide pool. Thus, the efficiency bias presents a constraint on the number of re-amplifications that a DNA data storage system may go through before the uneven coverage distribution either prevents successful decoding or necessitates higher physical coverage and sequencing depth[4,21].

## Quantifying errors during storage

The detrimental impact of long-term storage on DNA data storage systems is well established, and usually quantified by the loss of amplifiable DNA over time[5,36,37]. Here, in addition to quantifying this loss of DNA, we also tracked the evolution of errors and biases during rapid aging by sequencing the oligonucleotide pools at various storage durations, up to the equivalent of >1000 years at 10 °C (7 days at 70 °C, see Fig. 4a). We observed a linear increase in C→T and G→A transitions as the major type of substitution errors, with ~$1.64 \times 10^{-4}$ $nt^{-1}$ per half time of decay overall (see Fig. 4b, c). In addition, a small number of deletions were introduced. These were negligible compared to the deletions present due to the synthesis (see Supplementary Fig. 14). Overall, the measured error rates show that storage-induced decay is not a significant error source in the context of DNA data storage. Comparing to other error sources, storage for eight half-lives—equivalent to the loss of 99.6% of DNA—introduces less errors than just 15 cycles of standard, Taq-based PCR. Therefore, the main effect of storage-induced decay is limited to the loss of sequences, and we focussed on characterizing any possible bias in this loss.

To assess the overall bias in decay, we compared the coverage distributions between aged samples and an equally diluted and amplified, but unaged, reference. We observed no difference in the

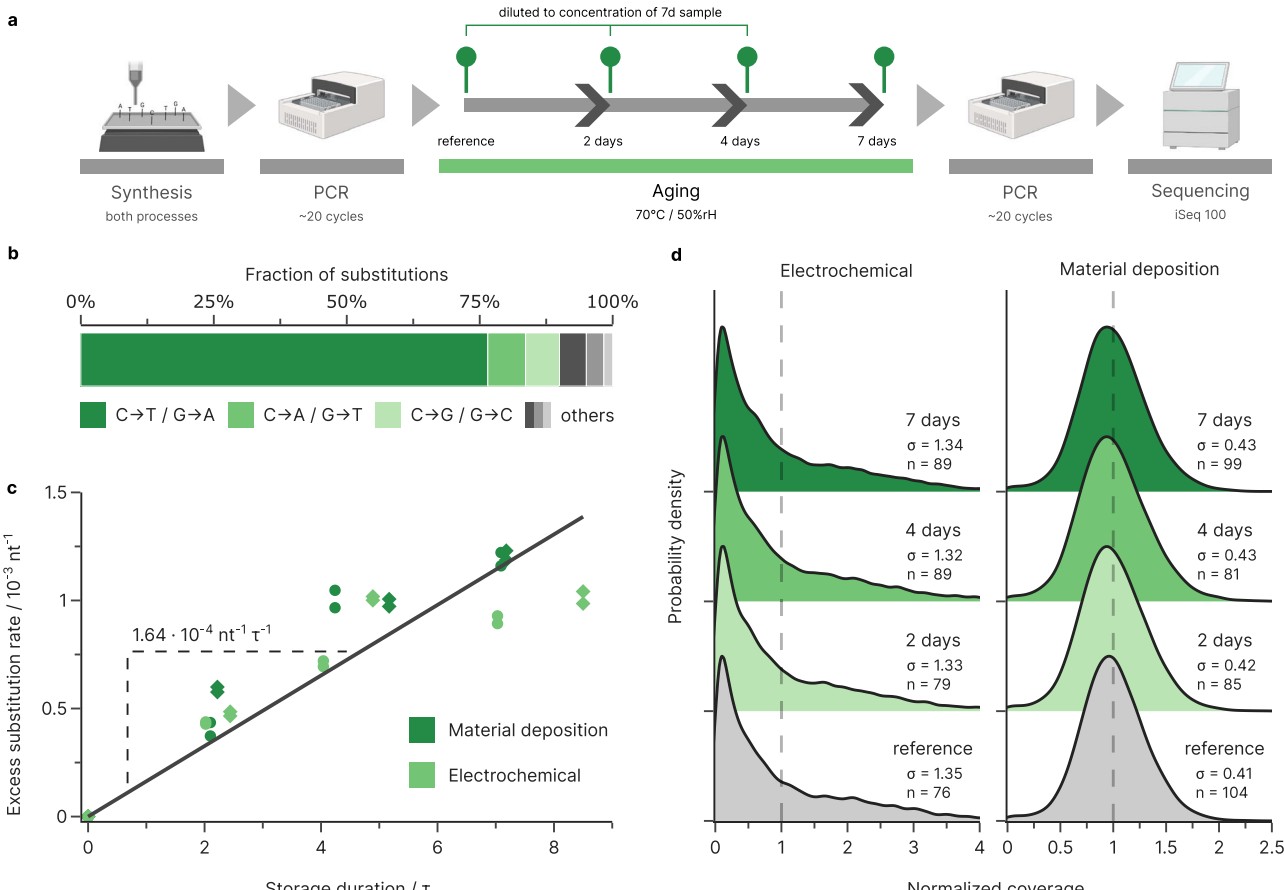

**Fig. 4 | Errors and biases during storage. a** Experimental workflow for estimating the error rates and biases during aging. **b, c** Substitutions introduced as a function of the total storage duration in half-lives, using the error rates of the unaged reference as baseline. Substitutions increase at a rate of $1.64 \times 10^{-4}$ per nucleotide per half-live ($nt^{-1} \tau^{-1}$) based on the regression slope (solid line). Substitutions are mainly C→T/G→A transitions (dark green, 77%) with minor C→A/G→T and C→G/G→C transversions (7% and 6%, respectively). **d** Kernel density estimate plot of the oligonucleotide coverage for the GC-unconstrained samples which were only diluted

(reference, gray), and samples that underwent decay for 2–7 days (green), for both electrochemical (left) and material deposition-based synthesis (right). All samples were diluted to the same concentration prior to amplification. The gray distribution shows the effect of subsampling via dilution, whereas the other distributions show the combined effects of dilution and decay. The standard deviations of the log-normalized distributions, $\sigma$, and the mean sequencing coverage, $n$, are given in the plot. Source data are provided as a Source Data file.

coverage of aged samples compared to unaged, but diluted samples (see Fig. 4d), meaning decay did not introduce considerable additional bias over random sampling. As such, the considerable difference in coverage bias between the two different synthesis processes remains dominant, as observable in Fig. 2b and discussed in the synthesis section above. Thus, the impact of decay on coverage distribution is well approximated by random sampling and any potential bias is likely secondary to the stochastic effects from sampling at low physical coverage. As aging neither introduced errors at relevant rates, nor significantly affected the coverage distribution in our experiments, recovered oligonucleotides (i.e., those without strand breaks induced by β-elimination) remained virtually unaffected by decay. This implies that long-term storage does not negatively impact the error resilience or fidelity, as long as sequence dropout is limited by sufficient coverage or enzymatic repair[36].

### Inhomogeneities in sequencing errors

We further investigated the errors introduced during Illumina sequencing by characterizing the error profile of reads mapped to PhiX, a common spike-in used as sequencing control and for color balancing. For our analysis, we consider PhiX—a PCR-free, adapter-ligated sample derived from genomic DNA[38]—essentially error-free and attribute all errors in its sequencing data to the sequencer. Using the eight PhiX datasets generated during sequencing on the Illumina iSeq

100 sequencer, we found substitutions are dominating, at $1.8 \pm 0.8 \times 10^{-3}$ $nt^{-1}$ on average, versus $<0.1 \times 10^{-3}$ $nt^{-1}$ for both deletions and insertions. This is in-line with other reports for other SBS-based sequencers[19,20,39] and the analysis of non-consensus errors between paired reads in our datasets (see Supplementary Fig. 15). The substitution rates in our experiments differed substantially between forward ($1.1 \pm 0.3 \times 10^{-3}$ $nt^{-1}$) and reverse reads ($2.5 \pm 0.6 \times 10^{-3}$ $nt^{-1}$), and were strongly cycle-dependent (see Fig. 5a). They declined rapidly towards a minimum around cycle 20, which coincides well with the calculations for phasing/pre-phasing and color-matrix corrections occurring at cycle 25[40]. After cycle 25, the number of substitutions slowly increased each cycle (see Fig. 5a).

The substitutions introduced during sequencing showed a clear bias towards base transitions (e.g., A↔G and C↔T) over transversions (all other combinations, see Fig. 5b), which differed slightly between forward and reverse reads. Moreover, the increase in substitution rate after cycle 20 appears to be primarily caused by A→T and T→G substitutions, while all other substitution patterns remain nearly constant throughout the duration of the sequencing run (see Supplementary Fig. 16). The comparison to the base-calling method used in the iSeq's one-dye sequencing (see Fig. 5b, inset) shows that base transitions correspond to false positive and false negative calls in the primary image, accounting for 54% of all sequencing errors on average. A major exception is the A→T transition, responsible for an additional $17 \pm 5\%$

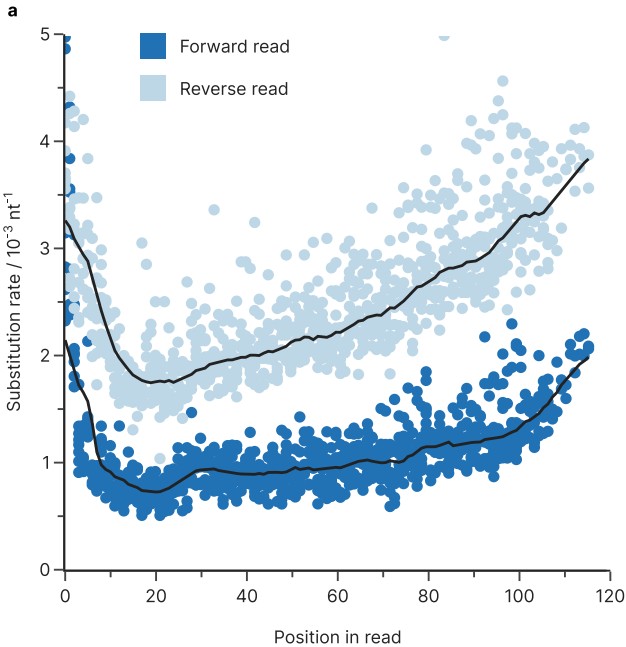

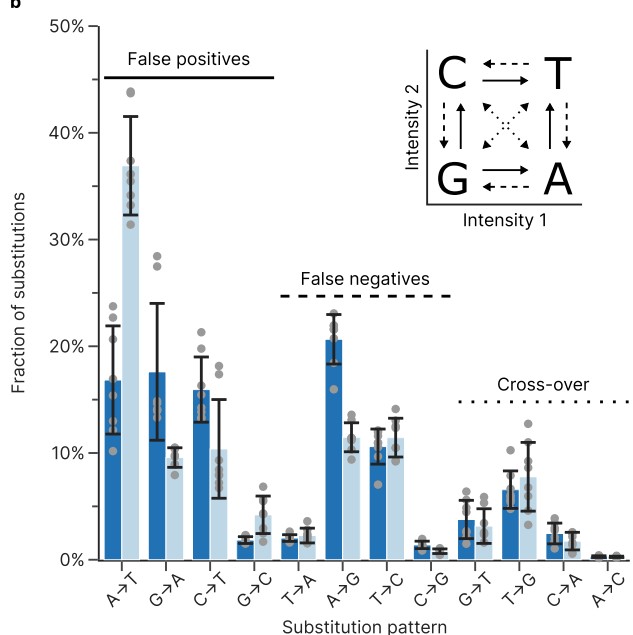

**Fig. 5 | Errors and biases from Illumina sequencing. a** Substitution rate during sequencing on the Illumina iSeq 100, estimated from the PhiX reads obtained during all eight sequencing runs. Points show the individual substitution rate of the forward (dark blue) and reverse reads (light blue) at every position, with their respective moving median (10 base window, black lines). Only the positions until cycle 112 are shown, as low base diversity in the priming regions of the co-sequenced oligonucleotides drastically skews base-calling accuracy. **b** Base bias of substitutions occurring during sequencing in the forward (dark blue) and reverse reads (light blue), shown as fractions of the total substitutions. The one-dye sequencing system used by the iSeq 100 sequencer (inset) uses the fluorescence intensity in two separate images for base calling[40]. Depending on which fluorescence signal is miscalled, false positive (solid), false negative (dashed), or cross-over (dotted) errors occur and introduce a substitution into the sequencing data. Colored bars show the mean and error bars the standard deviation of the sample (gray points, $n = 8$ sequencing experiments), grouped by read direction during sequencing. Source data are provided as a Source Data file.

and $37 \pm 5\%$ of substitutions in the forward and reverse reads respectively, which corresponds to a false positive in the secondary image. Thus, unlike for sequencers with other dye chemistries[20], substitution bias on the iSeq 100 appears to be related to its base-calling matrix. Underlining this, substitutions involving miscalling intensities in both images ("cross-over" in Fig. 5b) were rare and accounted for only 15% of substitution errors. Additionally, the analysis of non-consensus errors between paired reads in our datasets (see Supplementary Fig. 15) suggests that polymerase errors during clonal amplification (i.e., the clustering step in SBS) also skew the substitution bias.

## A digital twin for DNA data storage

Towards our goal of providing an accurate virtual representation of DNA data storage experiments, we implemented the error sources and biases characterized above into a digital twin of the DNA data storage process (see Fig. 6a). The digital twin's underlying model simulates all process steps (e.g., synthesis, PCR) by stochastically introducing mutations into sequences at rates estimated from user-supplied experimental parameters. Specifically, we represent an oligonucleotide pool as a collection of sequences with associated abundances and use many oligonucleotides for each sequence to accurately represent the experimentally observed diversity of error patterns. Importantly, the biases introduced into the coverage distributions by synthesis, amplification, and dilution are also modeled (e.g., by skewed initial distributions as in Fig. 2b, or non-homogeneous amplification as in Fig. 3d), so that their negative effects on coverage homogeneity and sequence dropout are included. Additional information and details on the implementation of each process step are given in the Methods and Supplementary Note 2.

To assess our model's accuracy and versatility in predicting errors and biases from an experimental workflow, we reproduced the experiments presented in this study (as internal validation) and modeled the generational experiments by Koch et al.[23] (as external validation). These generational experiments, starting from an electrochemically synthesized oligonucleotide pool, are ideal for model validation: they consist of multiple dilutions and error-prone re-amplifications—exceeding 100 PCR cycles in total—and include seven sequencing datasets for comparison. We observed good agreement in the overall error rates and the coverage bias for both internal ($R^2_{\text{error}} = 0.98$, $R^2_{\text{bias}} = 0.74$, see Supplementary Note 5) and external validation ($R^2_{\text{error}} = 0.87$, $R^2_{\text{bias}} = 0.64$, see Fig. 6b and Supplementary Note 5). Notably, the experimental deletion rates in the generational experiments by Koch et al.[23] exceeded the prediction of our model by about 20%, mostly due to differences in the position-dependent deletion rates during synthesis (see Supplementary Fig. 17). This difference is likely caused by the implementation of process improvements by the synthesis provider sometime between the study by Koch et al. and this work. This highlights the possible relevance of the digital twin for the investigation of process deviations. Turning to coverage bias, we considered the rate of sequence dropout—i.e., the ratio of original sequences which are no longer present in the sequencing data—as our metric, due to its relevance for successful data recovery in a data storage context. We found that our simulated sequencing data, downsampled to the original experiment's read counts, accurately reproduced the sequence dropout observed over all seven generations (see Fig. 6c). Importantly, had Koch et al.[23] been able to model their workflow, they would have been able to increase storage capacity (by reducing redundancy) or lower costs (by synthesizing fewer sequences) by more than threefold (the authors included redundancy for a sequence dropout of 80%, but a maximum of 30% was required). Alternatively, using the model to forecast future generations of Koch's experiment, at least four more generations would have been feasible at their redundancy level. This analysis highlights the value of the digital twin for the rational design

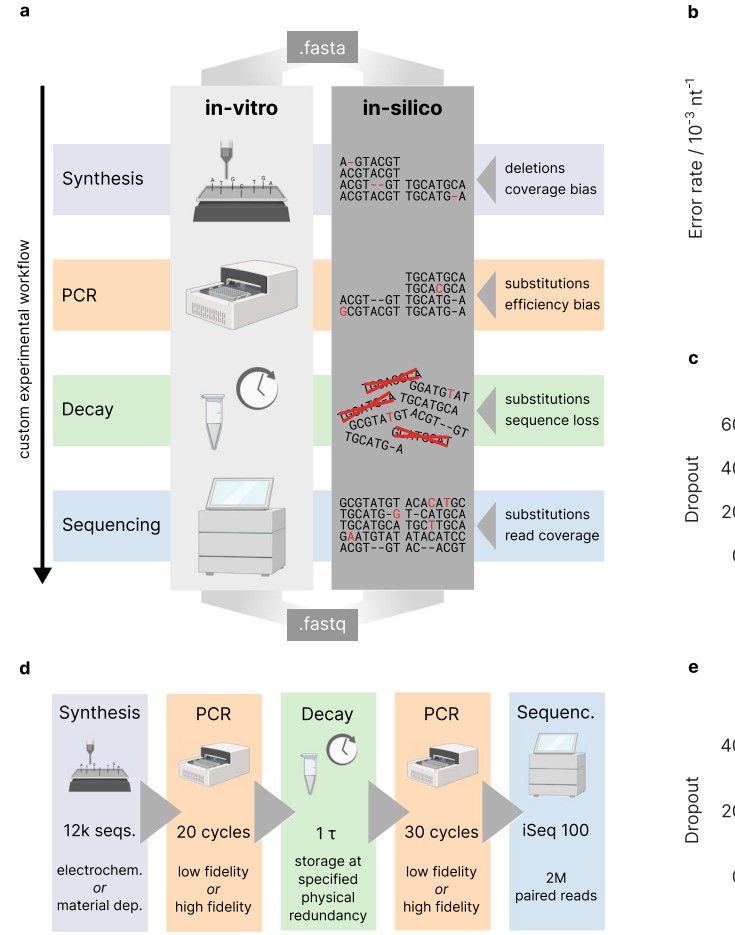

**Fig. 6 | Simulation of the DNA data storage channel. a** Overview of the developed model for the DNA data storage channel. Experimental parameters for the synthesis, amplification, decay, and sequencing are used to replicate errors and biases in an in-silico representation of an oligonucleotide pool. The order and parameters of all process steps can be customized to describe user-defined workflows. **b, c** Verification of simulation results using the generational experiments reported by Koch et al.[23]. The mean error rates (**b**) and sequence loss (**c**) of the data storage workflow, as experimentally observed (Exp., light gray) and as replicated in our model (Sim., dark gray), is shown for the master pool (denoted M), the parent (P), and all progeny generations (F1 through F5). The model was also used to predict four further generations (F6 through F9). Datapoints are slightly offset horizontally to prevent occlusion. Sequencing data from the model was downsampled to the read count in the experimental sequencing data. **d, e** Simulation of the effects of physical coverage on sequence dropout in a best- and worst-case scenario. By implementing a typical data storage workflow (**d**) using high- or low-fidelity process steps in our model, the sequence dropout (**e**) as a function of physical redundancy is determined. The loss of sequences considering both all sequencing reads (solid line) and only error-free reads (dashed line) is reported, with the shaded area in-between denoting the improvement possible by error correction coding. For comparison, the state-of-the-art storage density and redundancy by Organick et al.[22] is shown (black diamond, 6.2× coverage at 15% data redundancy). Source data are provided as a Source Data file.

of redundancy: it enables cost-saving optimizations and facilitates experimental planning.

## Case study: optimal redundancy in extreme scenarios

To highlight the value of modeling each process step for the design of redundancy in DNA data storage systems, we implemented a prototypical storage workflow in our model as a case study. To investigate optimal physical and logical redundancy, our prototypical workflow (see Fig. 6d)—involving post-synthesis amplification, dilution to a specified physical coverage, storage for one half-live, re-amplification, and sequencing—was further divided into two extreme cases. In our worst-case scenario, an unconstrained, electrochemically synthesized oligonucleotide pool was used (see Fig. 2b) together with a low-fidelity polymerase for PCR. Due to the highly skewed coverage and large error rate, this scenario is representative of studies in which high redundancy is favored and storage density is not the main concern[7,23,41]. In contrast, the best-case scenario utilized a narrowly distributed oligonucleotide pool synthesized by a material deposition-based process, and further used a high-fidelity polymerase for amplification. This is a

low-error, low-bias scenario like those used in many studies on ECC[6,10]. As expected, our model predicted that the physical redundancy used during storage, i.e., the effectively achieved storage density, strongly influences the sequence loss in both our scenarios (see Fig. 6e). The less biased best-case scenario yielded near-complete recovery (98%) of error-free sequences with only 10 copies per sequence during storage, corresponding to a storage density close to the experimentally demonstrated state-of-the-art (6.2× coverage, 15% redundancy)[21,22]. In contrast, the worst-case scenario lost 24% of all sequences at the same physical redundancy, highlighting the importance of coverage homogeneity for high-density DNA data storage.

Logical redundancy implemented into an ECC provides two main benefits: first, it tolerates the loss of a certain number of sequences (via redundant sequences); second, it enables the use and decoding of erroneous reads if no error-free reads of a sequence are available (via within-sequence redundancy). The latter benefit effectively yields either a gain in storage density or a gain in sequence coverage, as shown when moving from the curve considering only error-free reads (naive encoding, no within-sequence redundancy) to all reads (ideal

ECC, capable of decoding every erroneous read) in Fig. 6e. To take full advantage of this gain in density or coverage, an ECC would have to be able to correct up to two deletions and two substitutions per sequence in our low-fidelity scenario. However, our model shows that even just the capability to correct up to two substitutions would approximately double the number of eligible reads, as deletions are clustered in only 48% of reads (see Fig. 1d). In contrast, the implementation of such within-sequence error correction would prove wasteful in our high-fidelity scenario. There, considering only error-free reads does not significantly deteriorate sequence coverage, as 81% of reads are error-free on average anyway. Consequently, a naive encoding without within-sequence redundancy will achieve a higher storage density in the best-case scenario than any other ECC in the worst-case scenario, independently of the ECC's capabilities.

## Discussion

The lack of comprehensive data on error rates, error homogeneity, and coverage biases throughout the DNA data storage workflow has restricted users from rationally selecting redundancy levels and understanding the impact of workflow choices. In this work, we have comprehensively quantified errors and biases in DNA storage systems and developed a digital twin for modeling state-of-the-art data storage workflows. Systematic sequencing of oligonucleotide pools during processing showed that synthesis and standard PCR account for most deletions and substitutions, which outnumber insertions by a factor of >10. Deletions were almost exclusively introduced by synthesis and heterogeneously distributed in clusters. All other processing steps—amplification via PCR, aging, and sequencing by SBS—added substitutions at varying rates, which were homogeneously distributed but biased towards certain substitution patterns. Remarkably, the state-of-the-art data storage workflow has become close to error-free (up to 87% of forward reads without error, 96% deletion-free), as shown in our idealized high-fidelity storage scenario (see Fig. 6d). This implies some of the ongoing optimization of ECCs towards increased error resilience to be better suited for applications in which low-fidelity synthesis or sequencing processes require an ECC capable of utilizing highly erroneous reads[41,42]. In contrast, the commonly used workflow for high-density DNA data storage—based on synthesis via material deposition and high-fidelity PCR—does not appear to benefit from such ECC optimizations, as storage density is currently limited by coverage biases.

Synthesis and amplification also emerged as the major contributors to skewed coverage distributions in our systematic analysis of coverage bias in synthetic oligonucleotide pools. While unoptimized synthesis processes and the stochasticity of amplification are known to affect the coverage distribution[21], we identified both a striking difference in coverage uniformity between two different synthesis processes and an apparent bias in the amplification efficiency during PCR. The consideration of these coverage biases was shown to be crucial for understanding sequence dropout, a vital metric for error-free readout due its severe effect compared to single mutations—necessitating redundant sequences rather than just redundant symbols.

Our experimentally verified digital twin showcased the value of a customizable digital representation of the DNA data storage process for experimental planning and the ECC design. The digital twin facilitated the design of redundancy both in a literature scenario and a case study, which was shown to translate into tangible cost savings. Furthermore, it highlighted that sequence dropout caused by coverage bias, rather than erroneous sequences caused by mutations, is currently the limiting factor in designing DNA data storage systems with increasingly higher storage densities. To this end, novel approaches to remedy sequence dropout—such as ECCs capable of utilizing partial sequences[43] or methods for enzymatic DNA repair[36]—will be invaluable to facilitate long-term storage at these high storage densities.

Key limitations of our study include the consideration of only two commercial providers for synthesis and only Illumina's SBS technology for sequencing. While these technologies are currently the most relevant and widely used[1,7], other emerging technologies—such as photoarray-based or enzymatic synthesis, as well as nanopore sequencing—are expected to soon become relevant cost-effective alternatives despite their lower fidelity[3,41,42]. For this reason, our digital twin is modular and thus easily expandable, enabling the implementation of new processes, irrespective of their error rates and biases. Importantly, it also accommodates non-ideal error patterns, such as the burst deletions that were observed during synthesis (see Fig. 1b) and may arise in other processes, such as nanopore sequencing[44]. Furthermore, the broad scope of our analysis precluded a detailed investigation into individual error sources, such as the effects of different polymerases or correlations with sequence properties (e.g., GC content, homopolymers). Despite these limitations, we hope both our error characterization and our digital twin will help standardize the comparison and accelerate the development of ECCs, as well as assist users in designing redundancy and experimental workflows. For this, we provide a web platform to simulate both standardized and customized storage scenarios at dt4dds.ethz.ch, as well as source code for fully custom workflows at github.com/fml-ethz/dt4dds[45]. We also invite others to extend our model with more data, especially for the emerging, low-fidelity technologies previously mentioned.

## Methods

### Reagents

Electrochemically synthesized oligonucleotide pools were ordered from CustomArray Inc. (Redmond, WA, United States) and Genscript Biotech Corp. (Piscataway, NJ, United States) and used as delivered. Material deposition-based oligonucleotide pools were synthesized by Twist Bioscience (San Francisco, CA, United States) and resuspended to 10 ng μL$^{-1}$ in ultrapure water. Primers were purchased from Microsynth AG (Balgach, Switzerland). All pools and primers were further diluted as required with ultrapure water. Additional details about the design of oligonucleotide pools and primers are given in Supplementary Tables 1 and 2. KAPA SYBR FAST polymerase master mix was purchased from Sigma-Aldrich (St. Louis, MI, United States).

### PCR and sequencing preparation

Unless otherwise noted, 5 μL of an oligonucleotide pool and 1 μL each of the forward and reverse primers (0 F/0 R, 10 μM) were added to 10 μL of 2× KAPA SYBR FAST master mix. Ultrapure water was added up to a final volume of 20 μL. Amplification by PCR used an initial denaturation at 95 °C for 3 min, followed by cycles at 95 °C for 15 s, 54 °C for 30 s, and 72 °C for 30 s. Cycling was stopped as soon as the fluorescence intensity reached its plateau to prevent resource exhaustion, except for quantitative PCR (calibration curves are given in Supplementary Fig. 11). For sequencing preparation, indexed Illumina adapters were added by PCR with overhang primers (2FUF/2RIF, 7-9 cycles, see Supplementary Table 2). The PCR product from each well was then run on an agarose gel (E-Gel EX Agarose Gels 2%, Invitrogen) with a 50 bp ladder (Invitrogen), and the appropriate band was purified (ZymoClean Gel DNA Recovery Kit, ZymoResearch) before quantification by fluorescence (Qubit dsDNA HS Kit, Invitrogen)[7].

### Sequencing

For each sequencing run, 5–6 samples were individually diluted to 1 nM and pooled. The pooled sample was further diluted to 50 pM. Then, 2% PhiX (PhiX Control v3, Illumina) was spiked into the sample and 20 μL were added to an Illumina iSeq 100 i1 Reagent v2 cartridge. 150 nt paired-end sequencing with the Illumina iSeq 100 sequencer yielded between 4 and 5 million reads, leading to an average sequencing coverage of 90 paired reads per sequence.

## Protocol for amplification experiments

For the amplification experiments, oligonucleotide pools were sequentially amplified and diluted multiple times under the same conditions to yield samples at six different PCR cycle counts. For this, the pools synthesized by material deposition (500× dilution) were amplified in two wells each, one well containing standard primers (0 F/ 0 R) and one containing the indexed overhang primers with sequencing adapters (2FUF/2RIF). After 15 cycles, the PCR product with sequencing adapters was stored at −20 °C. 1 μL of the PCR product with the standard primers was diluted by 3800×, and 5 μL were used for the next round of amplification (for a total dilution of 15,200×, equivalent to $1.9^{15}$, the expected amplification factor after 15 PCR cycles with 90% efficiency). If the fluorescence observed in the last cycle of an amplification round was approaching the plateau value, the dilution for the next round was increased two-fold, i.e., to 7600×. This sequential procedure was performed for a total of six rounds, yielding samples with 15 to 90 PCR cycles. The PCR products with sequencing adapters were then prepared for sequencing (see above) without the additional indexing step. The workflow is shown in Supplementary Figs. 18 and 19.

The procedure and results for the amplification experiments of the electrochemically synthesized pools (not shown in Fig. 3) are given in Supplementary Note 4. The workflow is illustrated in Supplementary Figs. 20 and 21.

## Protocol for storage experiments

Both the electrochemically synthesized pools (50× dilution) and the pools synthesized by material deposition (1000× dilution) were first amplified for 20–21 cycles, using 96 wells each and 1 μL sample per well. Then, all wells from each pool were pooled and purified (DNA Clean & Concentrator-5, ZymoResearch) to yield stock solutions with 30–50 ng μL$^{-1}$ dsDNA in ultrapure water. Of these, 30 ng each were added to microcentrifuge tubes and dried in vacuo for 30 min at 45 °C. After drying, one set of tubes was immediately stored at −20 °C to represent the unaged reference sample. For accelerated aging, all other samples were stored in a desiccator over saturated sodium bromide in water (>99%, Roth AG) at 70 °C and 50% relative humidity[5]. Samples were moved to −20 °C storage after around two, four, and seven days, with each time point at least in triplicate. All samples were resuspended in 200 μL ultrapure water and quantified by qPCR to yield a decay curve, as described below. Calibration curves for this qPCR analysis were previously established by serial dilution of the stock solutions and are shown in Supplementary Fig. 11 with their parameters given in Supplementary Table 3. For the decay curve, the concentration of all samples was normalized to the mean concentration of the unaged reference sample, and then fitted to a first-order decay model according to Eq. 2.

$$\frac{c(t)}{c(0)} = e^{-kt}, \text{ where } k = \frac{\ln 2}{\tau}. \tag{2}$$

The decay curves and their parameters are given in Supplementary Fig. 11 and Supplementary Table 4, respectively.

For sequencing, all samples were diluted to the concentration of the sample at 7 days to circumvent any dilution effects, amplified for 16-18 cycles, and then underwent the standard sequencing preparation (see above). The workflow is shown in Supplementary Figs. 22–25. To normalize the extent of decay across the four oligonucleotide pools for the estimation of error rates during aging, the number of half-lives, determined as the storage duration relative to the half-live, was used. The conversion for all time points is given in Supplementary Table 5.

## Read mapping and error analysis

To estimate error rates from sequencing reads, up to 1 million paired-end sequencing reads were first mapped to their respective reference sequence using a custom Python script, and then filtered to exclude reads with <85% similarity to their reference. This filtering threshold was chosen based on similarity comparisons between experimental and random datasets (see Supplementary Fig. 1). From the resulting mappings, error rates as a function of position, involved bases, read direction, and error length were derived and used for further data analysis. Coverage distributions were derived from the alignment counts given by sequence alignment with BBMap[46] after adapter trimming and normalization to the mean oligonucleotide coverage. Lognormal distributions were fitted to the normalized coverage distributions to help with visualization, and the corresponding standard deviation of the lognormal distribution is shown to quantify the coverage bias. Full details are given in Supplementary Note 2 and the complete source code is publicly available in the GitHub repository (see Code Availability statement).

## ANOVA and error independence

Three-way ANOVA ($n = 80$) with the factors synthesis provider, number of PCR cycles, and days of storage was performed using type II sum of squares, heteroskedasticity-consistent standard errors (HC3), and without interactions. The analysis was performed for each error type independently and according to the following linear model in Equation 3.

$$\text{Error rate} \sim C(\text{synthesis}) + \text{PCR cycles} + \text{Days of storage} \tag{3}$$

For the analysis of error independence, theoretical probability mass functions under the assumption of error independence were independently calculated for each pool and experiment. For the probability mass function of consecutive errors, a geometric distribution parameterized by the mean error rate was used, i.e., $n \sim \text{Geom}(1 - \text{mean error rate})$. For the probability mass function of errors per read, a binomial distribution parameterized by the length of the sequence and the mean error rate was used, i.e., $n \sim \text{Binom}(\text{length, mean error rate})$.

## Modeling of the DNA data storage process

The model used for the simulation of the DNA data storage process, implemented in Python, consists of a hash map representing a pool of oligonucleotides, error generators introducing mutations at specified rates and with certain biases, and classes encapsulating the error generators into the individual process steps (i.e., synthesis, PCR, storage, and sequencing). Starting from a set of reference sequences and an experimental workflow provided by the user, the model simulates errors and biases and ultimately yields artificial sequencing data in the FASTQ format for further use. The individual error sources and coverage biases of each process step are reproduced based on user-defined experimental parameters (e.g., synthesis provider, choice of polymerase, storage duration) and the error rates and biases quantified in this study. Coverage bias is implemented both during synthesis —via skewed initial count distributions as in Fig. 2d—and during amplification, using normally distributed relative amplification efficiencies as in Fig. 3d. Additionally, amplification is implemented as a branching binomial process, based on oligonucleotide count and the sequence's amplification efficiency, to account for the stochastic effects observed at low coverage[21,27]. Dilution, sequencing, and decay are modeled as random sampling, in-line with the findings in Fig. 4 and the literature[4,21]. Full details are given in Supplementary Note 2 and the complete source code is publicly available in the GitHub repository (see Code Availability statement).

## Internal and external validation

For the internal validation, all experimental conditions from this study were recreated with our tool and the simulated sequencing data underwent identical post-processing and error analysis. Only the position-, length-, and base-dependent error rates, process-specific error patterns, and coverage biases characterized in this study were

utilized. Due to small differences in the positional deletion rates between the two electrochemically synthesized pools, pool-specific deletion rates were used (see Supplementary Note 3) rather than the overall deletion rate presented in Fig. 2a.

For the external validation, the workflow for the generational experiments by Koch et al.[23] was reproduced with our tool to the extent possible given the information provided in their study. Electrochemical synthesis was assumed with positional error rates as in Fig. 2a, and a coverage bias of $\sigma = 0.94$ (mean of GC-constrained and unconstrained pools, see Fig. 2b) due to their use of a partially GC-constraining ECC. Amplification by PCR assumed a Taq-based polymerase with an amplification bias as estimated for the Koch et al. experiments in Fig. 3d (i.e., $\sigma = 0.012$). Missing information about dilutions were estimated from other protocols[7] and the number of PCR cycles used. For the analysis in Fig. 6c, only error-free reads were used —as in the original study—and the simulated sequencing data was downsampled to the same read count as the experimental data to ensure comparability. For the generations F6-F9, the average read count of generations M-F5 was assumed.

More details on the parameters and results for both internal and external validation are presented in Supplementary Note 5. The scripts for both internal and external validation are also provided with the code in the repository for reproducibility.

### Case study on storage density
The best- and worst-case scenarios implemented in our tool were both based on the error characterization in this study and common experimental workflows for high-density DNA data storage[6,8,11,22]. The scenarios followed an identical workflow (see Fig. 6d and below) consisting of synthesis, amplification, storage, re-amplification, and sequencing. Specifically, 12000 sequences were synthesized at a mean coverage of 200, underwent 20 PCR cycles with an amplification bias of $\sigma = 0.0051$ (see Fig. 3c), were stored for one half-life at mean coverages ranging from 0.5-50 oligonucleotides per sequence, amplified for another 30 cycles, and finally sequenced with the iSeq 100. In the best-case scenario, the coverage bias and error rate of the material deposition-based synthesis (see Fig. 2), and the polymerase fidelity of Q5 High-Fidelity DNA Polymerase (i.e., 280)[18] were used. In the worst-case scenario, the coverage bias and error rate of electrochemical synthesis, and the fidelity of a Taq-based polymerase (i.e., 1) were used instead. For the analysis in Fig. 6e, either all or only error-free reads (see Supplementary Note 1) were used to determine the sequence dropout in both cases, equivalent to an ideal ECC, and a naïve ECC, respectively. The script for this case study is provided with the code in the repository for full documentation of the parameters.

### Reporting summary
Further information on research design is available in the Nature Portfolio Reporting Summary linked to this article.

## Data availability
The experimental and simulated sequencing data generated in this study have been deposited in the European Nucleotide Archive under accession code PRJEB65931. Sequencing data from the literature used for analysis is available from the studies by Koch et al.[23] (PRJEB35217), Erlich et al.[6] (PRJEB19305 and PRJEB19307), and Chen et al.[21] (github.com/uwmisl/storage-biasing-ncomms20). Source data are provided with this paper.

## Code availability
The code for error analysis and simulation of the DNA data storage process is deposited in the public GitHub repository at github.com/fml-ethz/dt4dds (https://doi.org/10.5281/zenodo.8329043)[45]. The code for data analysis, in the form of Jupyter Notebooks and data files, is deposited in the public GitHub repository at github.com/fml-ethz/dt4dds_notebooks (https://doi.org/10.5281/zenodo.8329037)[47].

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

## Acknowledgements

This project was financed by the European Union's Horizon 2020 Program, FET-Open: DNA-FAIRYLIGHTS, grant agreement no. 964995. We thank Dr. Max Horn and Dr. Philipp Antkowiak for the fruitful discussions that motivated this study, as well as their input on method development. Data analysis and simulations were performed on the Euler cluster operated by the High-Performance Computing group at ETH Zürich. Figures were partially created with BioRender.com.

## Author contributions

R.N.G. and R.H. initiated and supervised the project with input from W.J.S. A.L.G. performed the experiments, developed the code, performed data analysis, prepared illustrations, and wrote the manuscript with input and approval from all authors.

## Competing interests

The authors declare no competing interests.
