## [Peer Review File · Nature Communications]

Reviewers' Comments:

Reviewer #1:

Remarks to the Author:

This manuscript gives a well presented characterization of a "standard" DNA data storage work flow and presents a simulation model based on this characterization. I've read or reviewed several papers containing DNA storage simulators in the past, but to date none have contained careful characterization along with non-ideal models. This manuscript will stand out amongst its predecessors.

Minor issues:

fig 1 (and referencing paragraphs): I don't see it stated what the "theory" model is. I assume its exponential but its also unclear what parameter was chosen and why.

Citation 23: The first reference refers to the author as "Chen" while subsequent mentions, including in fig 3, refer to "Organick" (which appears to be reference 24 instead of 23). The authors should either correct the later appearances (eg line 229, line 256, and fig 3, and possibly more.).

line 213: ϵ_i is defined as "an individual sequence's efficiency" but not mathematically defined anywhere. At least to me this is unclear, and definitely not rigorously defined.

line 235: The meaning of "upper bound of the true amplification bias" is confusing. it sounds like your saying one should expect less bias than show in this work, but a process can always be made worse increasing the amplification bias. Is it meant to say lower bound? or is it referring to koch vs "this work" or something else?

line 250: "without PCR bias..." In the paragraphs on the previous pages the authors mention the stochastic effects also cause bias which seems to contradict this sentence. unless the authors statement was a tautology (eg "without bias, there is no bias").

figure 4d: the difference between the 0 day electrochemical and deposition distributions is stark. I don't see discussion on why. Were e-chem concentrations initially lower?

general: With only normalized data it's hard to determine if sample size effects caused stochastic dropout and skewing. It would be nice to see the unnormalized data somewhere or at least what the average coverage was in a lot of the figures.

Line 537: "Error! Reference source not found"

Supplemental note one: The definition of similarity given on page 1 uses the hamming distance. I suspect this is a typo and another metric was used as the use of hamming distance would string with a single deletion to likely not match, which would skew the data. I assume a different metric provided the python libraries was used, likely one that can handle gaps.

Reviewer #2:

Remarks to the Author:

This paper concerns DNA data storage, an exciting emerging technology which has the potential to revolutionize how we store data, due to its very high storage density and its stability. This paper makes two main contributions: (1) it characterizes errors across the DNA data storage pipeline, comprising DNA synthesis, PCR, aging, and sequencing; and (2) it builds a computational model of the workflow, which allows cost-effective optimization of error correction coding.

While the contents of the paper are interesting, valuable, and written well, I believe that it has limited scope for broad impact. Arguably, the main bottlenecks to making DNA data storage viable

are scalability and cost, with improvements in DNA synthesis and automated microfluidics being most pressing. In my opinion, an understanding of errors across the current DNA data storage pipelines is at a pretty good level already, and so are error correcting codes for DNA data storage. Of course, these can be optimized further, and this paper very much helps with this, but I think that we are at a point where we are seeing diminishing returns in this direction. For these reasons, I don't think the results will be of interest to the broad readership of Nature Communications; even among those who are interested in DNA data storage, interest in this paper will primarily come from a subset of this community who focus on error correction.

Furthermore, while the analysis in the paper is thorough and useful, it does have some serious limitations. A fundamental question, which is difficult to address, is: will this analysis still be relevant 5 or 10 years from now, as DNA data storage methods evolve? Arguably, DNA synthesis methods are very much still evolving, and in order for DNA data storage to become viable, DNA synthesis methods have to improve. Thus, quite possibly, the relevant DNA synthesis methods in 5-10 years will be different, and thus not addressed here. While this is somewhat unfair criticism, as the authors cannot really address this at the moment, similar arguments also hold for DNA sequencing (which is much more developed), and here the paper does fall short. Specifically, Nanopore sequencing has been explored in several DNA data storage systems, and this has advantages over Illumina sequencing in terms of cost and allowing longer sequences. However, Nanopore sequencing is also much more challenging in terms of errors, with a much higher error rate, and far from independent errors (e.g., bursty deletions). The authors do not study Nanopore sequencing (and I do not think that the developed computational model would fit Nanopore sequencing), and only briefly discuss this as a limitation in the Discussion section. Of course, it is fine to focus the study on workflows that use Illumina sequencing (since many do), but it weakens the authors' claim that their analysis is comprehensive.

Despite these criticisms, I think this is a well-written, valuable paper. Thus, while it is not suitable for publication in Nature Communications, it deserves to appear in a good specialized venue.

Response to Reviewers

Reviewer comments in *black*, author replies in *red* with actions in **bold**.
Line numbers refer to the revised manuscript.

Comments by Reviewer #1

This manuscript gives a well presented characterization of a "standard" DNA data storage work flow and presents a simulation model based on this characterization. I've read or reviewed several papers containing DNA storage simulators in the past, but to date none have contained careful characterization along with non-ideal models. This manuscript will stand out amongst it's predecessors.

We thank the reviewer for their work in reviewing our manuscript, and for recognizing the value of our characterization and model. Incorporating their comments has considerably increased the readability and quality of the manuscript.

Minor issues:

fig 1 (and referencing paragraphs): I don't see it stated what the "theory" model is. I assume its exponential but its also unclear what parameter was chosen and why.

The assumptions underpinning the distributions labelled as "theory" are only explained in the main text (lines 119ff). We agree with the reviewer that the omission of an explanation in the caption reduces the comprehension of the figure. **We have added an explanation to the caption of Figure 1.**

Citation 23: The first reference refers to the author as "Chen" while subsequent mentions, including in fig 3, refer to "Organick" (which appears to be reference 24 instead of 23). The authors should either correct the later appearances (eg line 229, line 256, and fig 3, and possibly more.).

We thank the reviewer for catching this mistake in the manuscript. **All mentions of these references were checked and corrected where necessary.**

line 213: ϵ_i is defined as "an individual sequence's efficiency" but not mathematically defined anywhere. At least to me this is unclear, and definitely not rigorously defined.

The reviewer raises an important point about the unclear definition of ϵ_i , i.e. an individual sequence's efficiency. In our model, this efficiency is the expected number of copies each oligo of the sequence i produces in each PCR cycle (i.e., the probability of successful amplification). For example, 100 oligos of a sequence with $\epsilon_i=0.92$ are expected to yield 92 copies (i.e., 8 oligos fail to amplify), for a total of 192 oligos after the PCR cycle. **A definition was added to the manuscript (lines 194ff).**

line 235: The meaning of "upper bound of the true amplification bias" is confusing. it sounds like your saying one should expect less bias than show in this work, but a process can always be made worse increasing the amplification bias. Is it meant to say lower bound? or is it referring to koch vs "this work" or something else?

The reviewer is correct in noting that the amplification bias can be worsened arbitrarily, compared to the amplification bias shown in this work. However, the sentence with the “upper bound of the true amplification bias” is intended to highlight that there are confounding experimental factors (such as those listed in lines 205-212) which might affect the observed bias in our analysis. As these confounding factors are all expected to increase the bias, the bias caused directly by the amplification in these experiments (“the true amplification bias”) must be equal or less than the observed bias. Thus, the observed bias shown in this work is an upper bound on the amplification bias, specific to those experiments, which represent a broad range of typical DNA data storage workflows. **We have clarified the section describing the upper bound on the amplification bias (lines 217ff) within our experimental scope.**

line 250: “without PCR bias...” In the paragraphs on the previous pages the authors mention the stochastic effects also cause bias which seems to contradict this sentence. unless the authors statement was a tautology (eg “without bias, there is no bias”).

We agree with the reviewer that the phrase is ambiguous, due to the possibility of stochastic effects. In the case of (old) line 250, we did not consider stochastic effects because the experiments of this work were conducted at high physical coverages. However, some literature experiments shown in the same figure likely include stochastic effects (we argue that this is partly why our estimate of the amplification bias is an upper bound, see lines 207-215 and the previous reply). **Thus, we have reformulated the argument (now lines 724-725 as this is a figure caption).**

figure 4d: the difference between the 0 day electrochemical and deposition distributions is stark. I don't see discussion on why. Were e-chem concentrations initially lower?

The distinct difference between the pools synthesized by electrochemical and deposition processes in Figure 4d mentioned by the reviewer is a result of the synthesis process. As discussed in the characterization of synthesis (lines 158ff) and illustrated in Figure 2b, the electrochemically synthesized pool has a highly skewed initial distribution compared to those synthesized by material deposition. This skew in the coverage is retained during storage. The effect might appear stronger than in Figure 2b due to the compression along the x-axis in Figure 4d. The initial concentration of the electrochemical pool after synthesis was indeed lower (see Supplementary Table 7), but we can assure the reviewer that this was accounted for and that the concentrations were similar between pools throughout the storage experiments. **We have added an additional sentence highlighting the cause of this difference between pools to the characterization of storage, referring to the section about synthesis (lines 246ff).**

general: With only normalized data it's hard to determine if sample size effects caused stochastic dropout and skewing. It would be nice to see the unnormalized data somewhere or at least what the average coverage was in a lot of the figures.

We understand the concern raised by the reviewer regarding sample size effects. We can assure the reviewer that sample size effects did not lead to significant stochastic dropout and skewing, as the pools were handled at high physical redundancies (>1000x) and sequenced using a sufficiently high sequencing coverage (average around 90x, minimum 57x). The only exceptions are the amplification experiments using the electrochemical pool discussed in the Supplementary Information (Supplementary Note 4), for which high dilution led to low physical redundancy and the stochastic

effects mentioned by the reviewer were observed. These experiments were only used to estimate error rates (see Supplementary Note 4), but not for the quantification of bias. Thus, we do not expect that stochastic effects play a role in our conclusions.

We have added information about the average sequencing coverage to each figure in the main text that shows normalized coverage data. Moreover, we added a Supplementary File containing relevant data (sequencing coverage, number of reads used for analysis, ...) for all experiments and added a sentence to the main text that mentions the minimum physical coverage and minimum sequencing coverage (lines 215ff).

Line 537: ``Error! Reference source not found``

Thanks for pointing out this mistake. This error is not present in the manuscript version that we prepared and may have been inadvertently introduced during the submission process. **We will double-check the revised version online prior to submission.**

Supplemental note one: The definition of similarity given on page 1 uses the hamming distance. I suspect this is a typo and another metric was used as the use of hamming distance would string with a single deletion to likely not match, which would skew the data. I assume a different metric provided the python libraries was used, likely one that can handle gaps.

We thank the reviewer for pointing out this mistake. This is indeed a typo, as the similarity metric used the Levenshtein distance. **“Hamming” was changed to “Levenshtein”.**

Comments by Reviewer #2:

This paper concerns DNA data storage, an exciting emerging technology which has the potential to revolutionize how we store data, due to its very high storage density and its stability. This paper makes two main contributions: (1) it characterizes errors across the DNA data storage pipeline, comprising DNA synthesis, PCR, aging, and sequencing; and (2) it builds a computational model of the workflow, which allows cost-effective optimization of error correction coding.

We thank the reviewer for their work in reviewing our manuscript and their constructive comments about the relevance of our work. Their comments prompted us to further clarify the scope of our work and highlight the expandability of our model.

While the contents of the paper are interesting, valuable, and written well, I believe that it has limited scope for broad impact. Arguably, the main bottlenecks to making DNA data storage viable are scalability and cost, with improvements in DNA synthesis and automated microfluidics being most pressing. In my opinion, an understanding of errors across the current DNA data storage pipelines is at a pretty good level already, and so are error correcting codes for DNA data storage. Of course, these can be optimized further, and this paper very much helps with this, but I think that we are at a point where we are seeing diminishing returns in this direction. For these reasons, I don't think the results will be of interest to the broad readership of Nature Communications; even among those who

are interested in DNA data storage, interest in this paper will primarily come from a subset of this community who focus on error correction.

We appreciate the positive feedback on the quality and value of our work by the reviewer. We agree with the reviewer that scalability and cost are currently limiting the viability of DNA data storage in practice, rather than the optimization of error correction codes. Moreover, as pointed out both by the reviewer and in our introduction (lines 55ff), many of the methods employed in DNA data storage have been characterized in isolation, and many error correction codes already exist. However, we believe this does not limit the usefulness of our work to the optimization of error correction codes. In fact, the aforementioned cost of data storage experiments and the broad availability of error correction codes are driving the significance of our contribution beyond the subset of the community concerned with developing error correction codes. When designing a DNA data storage medium, both the choice of error correction code and the selection of its parameters depend heavily on the anticipated errors and bias. Currently, balancing the errors and biases incurred by different synthesis processes or storage scenarios with the capabilities of error correction codes is challenging for the experimental community. Utilizing the results of our characterization and our model, it is easier to test and verify custom workflows and error correction codes prior to committing to a costly synthesis.

To underline the value of our work to the broader DNA data storage community, the introduction (lines 38ff) and discussion (lines 360-361) sections were rephrased.

Furthermore, while the analysis in the paper is thorough and useful, it does have some serious limitations. A fundamental question, which is difficult to address, is: will this analysis still be relevant 5 or 10 years from now, as DNA data storage methods evolve? Arguably, DNA synthesis methods are very much still evolving, and in order for DNA data storage to become viable, DNA synthesis methods have to improve. Thus, quite possibly, the relevant DNA synthesis methods in 5-10 years will be different, and thus not addressed here.

The reviewer has carefully highlighted the limitations of our work. We agree that DNA data storage methods in general, including DNA synthesis, will undoubtedly continue to evolve in the future. However, as also discussed by the reviewer, it is currently difficult to predict which synthesis method will be the most relevant for DNA data storage in the future. We would like to emphasize here that our work is not intended to characterize the errors and biases of all methods that are being used, or may be used in the future, for DNA data storage. Instead, we aim to comprehensively characterize and model the current state-of-the-art workflows, while our tools provide a solid foundation for future investigations into novel processes and workflows. **The parts of the introduction discussing the scope of our work were rephrased (lines 78ff) to highlight this.**

To this end, we would like to highlight that the adoption of novel methods to DNA data storage, such as new synthesis processes, does not render our analysis and modelling tools obsolete. Our analysis tool accommodates other synthesis or sequencing processes with little to no adaptations (e.g., it is used for the characterisation of errors in photochemical DNA synthesis and nanopore sequencing in DOI:10.48550/arXiv.2308.05952), and our model can be expanded to include such processes if information on their error rates and biases is available.

While this is somewhat unfair criticism, as the authors cannot really address this at the moment, similar arguments also hold for DNA sequencing (which is much more developed), and here the paper does fall short. Specifically, Nanopore sequencing has been explored in several DNA data storage

systems, and this has advantages over Illumina sequencing in terms of cost and allowing longer sequences. However, Nanopore sequencing is also much more challenging in terms of errors, with a much higher error rate, and far from independent errors (e.g., bursty deletions). The authors do not study Nanopore sequencing (and I do not think that the developed computational model would fit Nanopore sequencing), and only briefly discuss this as a limitation in the Discussion section. Of course, it is fine to focus the study on workflows that use Illumina sequencing (since many do), but it weakens the authors' claim that their analysis is comprehensive.

We agree with the reviewer that the omission of nanopore sequencing is a big limitation of our work, as it has seen increased use in the DNA data storage community. In addition to the reasons presented in the reply to the previous comment, in which similar concerns for DNA synthesis were raised, we refrained from considering nanopore sequencing because several sequencing workflows with different error profiles and limitations have been proposed and used (e.g. DOI: 10.1038/s41467-019-10978-4, DOI:10.1038/s41598-017-05188-1, DOI:10.1038/nbt.4079). As none of the workflows is clearly established, and the data sizes being read are significantly smaller than for Illumina-based sequencing, we feel that our omission of nanopore sequencing is justified, given the broad scope our study already has.

The reviewer also raises the valid point that nanopore sequencing exhibits higher error rates with other error patterns than Illumina-based sequencing. As shown in the characterization of electrochemical synthesis in the manuscript (lines 116ff), our model can account for the non-ideality of errors (e.g. bursty deletions). It does so by using empirical distributions for the frequency of errors per read and their length, rather than assuming that they are distributed independently. Thus, we can reassure the reviewer that it is feasible to extend our model to include nanopore sequencing workflows, given that information on error rates and biases is available. However, as outlined above, we consider this to be outside the scope of this work.

The part of the discussion addressing this limitation has been expanded (lines 398ff) to provide more perspective.

Despite these criticisms, I think this is a well-written, valuable paper. Thus, while it is not suitable for publication in Nature Communications, it deserves to appear in a good specialized venue.

We thank the reviewer for their constructive criticism of our work. Their comments have improved the manuscript and allowed us to highlight the expandability of our model. We hope these changes convince the reviewer of the value our work brings to the broader DNA data storage community.